

# Malignant solid tumor-related spontaneous intracerebral hemorrhage: a propensity score matching study

Shuolin Liang, Liuyu Liu, Bin Qin, Shengri Chen and Zhijian Liang

Department of Neurology, The First Affiliated Hospital of Guangxi Medical University, Nanning, Guangxi, China

Corresponding author
Zhijian Liang,
liangzhijian@gxmu.edu.cn

## ABSTRACT

**Background:** Spontaneous intracerebral hemorrhage (sICH) is a severe event with high mortality. Recently, evidence has emerged suggesting that malignant solid tumors may increase the risk of sICH through unique cancer-related factors. However, the specific risk factors and clinical characteristics of sICH in patients with malignant solid tumor remain poorly understood.

**Objective:** This study aimed to investigate the clinical characteristics of and identify the risk factors associated with sICH in individuals with malignant solid tumors.

**Methods:** This retrospective study was carried out in patients with active malignant solid tumors and sICH at the First Affiliated Hospital of Guangxi Medical University between January 2010 and December 2020. Patients were separated into control and malignant solid tumor-related spontaneous intracerebral hemorrhage (MST-sICH) groups. The control group consisted of patients presenting with malignant solid tumors alone who were matched to the MST-sICH group using a 1:1 propensity score matching (PSM) approach. Patient clinical data, laboratory findings, and imaging results were collected. Univariate analysis was carried out to determine the risk factors associated with MST-sICH. In addition, a receiver operator characteristic analysis was performed to identify potential predictors for poor prognosis.

**Results:** Decreased hemoglobin (HGB) levels, together with increased lymphocyte counts (LYCs), and an increased neutrophil-to-lymphocyte ratio (NLR) were found in the MST-sICH group compared to the control group. The results of the multivariate logistic regression analysis indicated a HGB levels (OR: 0.959, 95% CI [0.928–0.992]), an increased in LYCs (OR: 0.095, 95% CI [0.023–0.392]). Furthermore, there was an increased in NLR levels (OR: 2.137, 95% CI [1.427–3.200]). In the receiver operating characteristic (ROC) curve analysis, the area under the curve (AUC) with HGB, LYCs, and NLR as joint predictors was 0.955 (95% CI [0.901–1.000]), with a sensitivity of 100%, a specificity of 82.6%, and a Youden Index of 0.826.

**Conclusion:** Decreased HGB levels, elevated LYCs, and a higher NLR were identified as independent risk factors for sICH in patients with active solid malignancies. These markers could assist clinicians in stratifying high-risk patients, facilitating closer monitoring and informing targeted preventive strategies to mitigate the incidence of sICH in this at-risk population.

## INTRODUCTION

Spontaneous intracerebral hemorrhage (sICH) represents a severe cerebrovascular event with high rates of morbidity and mortality, particularly among cancer patients, where it presents distinctive challenges and risks. Although traditional risk factors such as hypertension are recognised as contributors to sICH, recent studies have indicated that malignancy itself significantly elevates the risk of this adverse event (*van Asch et al., 2010*; *McCormick & Rosenfield, 1973*; *Schrader et al., 2000*). This risk may be due to cancer-specific mechanisms, including coagulation abnormalities, systemic inflammation, and tumor-related vascular changes (*Iyengar et al., 2016*; *Bauer et al., 2022*; *Kim, Khorana & McCrae, 2020*). The prevalence of cancer in individuals experiencing sICH ranges from 1% to 10% (*McCormick & Rosenfield, 1973*; *Schrader et al., 2000*). A comprehensive nationwide survey involving approximately 820,000 individuals diagnosed with cancer revealed that the overall risk of hemorrhagic and ischemic stroke within 6 months of diagnosis was 2.2 and 1.6, respectively, indicating a heightened susceptibility to sICH in individuals with cancer (*Zöller et al., 2012*). Furthermore, in patients with systemic cancer, symptomatic bleeding was found to be more common than symptomatic infarction (*Graus, Rogers & Posner, 1985*). However, while previous studies have documented the association between cancer and sICH risk, they have primarily focused on cases with either hematological malignancies or brain metastases (*Chou & Singhal, 2007*; *Chen et al., 2012*; *Kewan et al., 2019*; *Srivastava et al., 2009*). These studies focus on factors such as coagulation abnormalities often seen in hematological cancers, or direct tumor effects in cases with brain metastases, but neglect to address the mechanisms underlying sICH in patients with solid tumors without these conditions. Another systematic retrospective study indicated that coagulation abnormalities and intratumoral bleeding were the main factors leading to sICH in patients with active malignancies, while traditional risk factors like hypertension (HTN) were less common (*Navi et al., 2010*). The leading factors contributing to sICH in cancer patients often include coagulation abnormalities associated with hematological malignancies, as well as brain metastases resulting from solid tumors. To date, studies have indicated that patients with solid malignant tumors may still be susceptible to sICH, even in the absence of intracranial metastases. The underlying mechanisms and specific risk factors contributing to sICH in non-brain metastatic cancer patients are still poorly understood, representing a crucial gap in current knowledge. Earlier studies placed greater emphasis on determining the differences between cancer patients with ICH and those with typical ICH, as this could identify latent malignancies in ICH patients (*Zhao et al., 2021*). The role of cancer as a prognostic factor in individuals suffering from ICH has also been examined (*Gon et al., 2018*). However, the etiology of sICH in patients with solid tumors in the absence of intracranial metastases or coagulopathy has not been fully explored. The presence of sICH in cancer patients is associated with a poorer prognosis and reduced overall survival. Given the severe

prognosis and unique clinical complexity of sICH in cancer patients, identification of specific risk factors is essential to improve risk stratification and inform preventive strategies in this high-risk population. By focusing on patients with non-brain metastatic solid tumors and explicitly excluding those with known coagulation disorders, our study aims to directly address the limitations of previous research and provide new insights into the risk factors associated with sICH in these patients.

The primary aim of this study was to retrospectively analyze data on patients with malignant solid tumors who developed acute sICH (MST-sICH patients), explicitly excluding those with brain metastases and coagulation dysfunction. Propensity score matching (PSM) analysis was employed to compare the clinical features and risk factors of MST-sICH patients against those with solid malignancies alone. A comprehensive range of clinical variables and demographic factors was examined to further our understanding of the intricate relationship between malignant solid tumors and acute sICH. Our findings aim to address a critical knowledge gap, with potential implications for improving clinical management and risk assessment strategies in cancer patients at risk of sICH.

## MATERIALS AND METHODS

### Patients

Data on patients with solid malignancies, with or without sICH, were collected from the First Affiliated Hospital of Guangxi Medical University (Guangxi Province, China) between January 1, 2010 and December 31, 2020. Individual clinical data included age, sex, smoking status, and alcohol consumption. High-risk factors for cerebrovascular disease included HTN, Type 2 diabetes, chronic heart disease and deep vein thrombosis, as well as the use of anticoagulants and antiplatelet medications. Routine blood examination parameters included white blood cell, red blood cell (RBC), and platelet counts, together with hepatic and kidney function tests and coagulation tests. Radiologic imaging included computed tomography (CT) and magnetic resonance imaging (MRI). Cerebrovascular imaging included CT angiography (CTA) and MR angiography (MRA). Imaging examinations were conducted within a single day and independently evaluated by at least two specialized neuroimaging physicians. Oncological data included types of malignancy, metastatic sites and pathological types. Other data were collected including treatment modalities, clinical symptoms and hematoma location. The modified Rankin Scale (mRS) was used to assess the degree of disability in patients with sICH. A mRS score ≤ 2 indicated a favorable prognosis and a score > 2 suggested an unfavorable prognosis.

The inclusion criteria for the MST-sICH group were: (1) individuals aged 18 years or older; (2) confirmation of sICH through CT or MRI; and (3) determination of malignant solid tumor types based on pathological diagnosis. The inclusion criteria for the control group were: (1) malignant solid tumor types based on pathological diagnosis; and (2) absence of sICH. The exclusion criteria for both groups were: (1) individuals below the age of 18: due to differences in tumor biology and risk profiles, pediatric patients were excluded; (2) incomplete medical records: such cases are defined as those which are missing key clinical data, such as imaging or laboratory results, in order to ensure the integrity of the data in question; (3) presence of liquid malignancy: liquid malignancies

(*e.g.*, leukaemia, lymphoma) were excluded as they increase the risk of bleeding through different mechanisms than solid tumors; (4) occurrence of subarachnoid hemorrhage, subdural and extradural hemorrhage: these types of hemorrhage differ from sICH in etiology and clinical management, which could confound our analysis; (5) presence of intracranial metastasis: excluded to specifically examine the effect of non-brain metastatic solid tumors on sICH risk; (6) presence of arteriovenous malformation, Moyamoya disease,intracranial venous thrombosis (CVST): these vascular conditions were excluded to avoid confounding, as they independently increase the risk of hemorrhage; and (9) abnormal coagulation function: coagulation disorders are independent risk factors for hemorrhage; excluding these cases allowed for a clearer assessment of cancer-specific sICH risks. Patient prognosis was evaluated using mRS scores, and patients were separated into favorable or unfavorable outcome groups based on their mRS score.

The article was examined by the Medical Ethics Committee of the First Affiliated Hospital of Guangxi Medical University, and the experimental scheme met the requirements of medical ethics (No. 2023-E246-01). The research project has been approved agree to exempt the research participants from additional informed consent of the project.

## Statistical analysis

Statistical analyses were conducted using IBM SPSS version 26.0. The Kolmogorov-Smirnov test was employed to determine the normal distribution of variables. Normally distributed variables were expressed as mean ± standard deviation, non-normally distributed variables were expressed as median (20–75%), and categorical variables were expressed as frequency (percentage). The Mann-Whitney U-test was employed to examine variables exhibiting non-normal distributions. Categorical variables were compared using Pearson's, chi-square, or Fisher's exact tests, while paired and unpaired t-tests were employed to assess disparities between continuous variables. A propensity score matching (PSM) analysis was conducted to mitigate the influence of conventional vascular risk on sICH in tumor patients. PSM analysis employed a multivariate logistic regression model that incorporated age, HTN, Type 2 diabetes, smoking status, and alcohol consumption. The PSM model included age, HTN, Type 2 diabetes, smoking status, and alcohol consumption, as these variables have well-established associations with cerebrovascular disease outcomes. To maintain a focus on baseline vascular risk factors, potential confounders such as anticoagulant use and metastatic status were excluded, as these may independently influence sICH risk. A total of 23 pairs of malignancy cases, with or without sICH, were matched using a 1:1 greedy nearest-neighbor approach based on propensity scores within a 0.02 range. Following matching, balance diagnostics, including standardised mean differences (SMD), were conducted to evaluate the effectiveness of PSM in achieving comparable groups on key covariates. SMD values below 0.1 were considered indicative of adequate balance. To analyze the non-normally distributed matched data, a Wilcoxon rank-sum test was utilized (Wilcoxon signed rank-test for comparisons within

groups). All *p*-values were two-sided, and a significance level of *p* < 0.05 was deemed statistically significant. It should be noted that the sample size could not be determined based on solid malignancy-related sICH features.

## RESULTS

### Demographic and clinical characteristics

A total of 836 patients with malignant solid tumors experienced sICH upon admission or during hospitalization. Among these patients, 26 individuals met all the inclusion criteria and were selected as the MST-sICH group, while 104 patients with solid malignant tumors alone acted as the control group. The average age of the MST-sICH patients was 61.8 ± 13.6 years compared to 56.7 ± 12.3 years in the control group (*p* = 0.061). No statistically significant differences in HTN, Type 2 diabetes, chronic heart disease, deep vein thrombosis, smoking status, and alcohol consumption were observed between the MST-sICH and control groups (38.5% *vs.* 20.2%, *p* = 0.051; 7.7% *vs.* 2.9%, *p* = 0.254; 34.6% *vs.* 32.7%, *p* = 0.852; 38.5% *vs.* 26.9%, *p* = 0.250). The most frequently observed clinical symptoms were coma (38.5%), nausea (30.8%), paralysis (34.6%), headache (23.1%), aphasia (15.4%), and sensory deficits (3.8%) (Fig. 1 and Table 1).

### Clinical data of the MST-sICH group

The most frequent primary sites of malignant tumors in MST-sICH patients were the lung (30.8%), liver (23.1%), intestine (23.1%), stomach (7.7%), bile duct (3.8%), breast (3.8%), adrenal glands (3.8%), and prostate (3.8%). The predominant pathological types of tumor were adenocarcinoma (61.5%), hepatocellular carcinoma (19.2%), squamous cell carcinoma (11.5%), and small cell lung cancer (7.7%) (Table 1).

### Imaging data

Following the onset of clinical symptoms, all patients underwent CT or MRI examinations. Traditional risk factors for sICH and intracranial tumor metastasis were excluded through comprehensive cerebrovascular imaging evaluations. The diagnosis of sICH was confirmed by either CT, which identifies acute hemorrhage as hyperdense areas within the brain parenchyma, or by MRI, where acute hemorrhage appears as characteristic signal changes on T1- and T2-weighted images. Moreover, each patient received at least one type of cerebrovascular imaging examination, including magnetic resonance angiography (MRA), magnetic resonance venography (MRV), computed tomography angiography (CTA), computed tomography venography (CTV), or digital subtraction angiography (DSA) was employed to conduct a comprehensive evaluation of cerebrovascular anatomy and to exclude traditional risk factors for sICH, including arteriovenous malformations, Moyamoya disease, and intracranial venous thrombosis, as well as intracranial tumour metastasis. The predominant sites of hemorrhage included the parietal lobe (50.0%), frontal lobe (26.9%), temporal lobe (19.2%), occipital lobe (19.2%), basal ganglia (19.2%), thalamus (19.2%), and cerebellum (15.4%) (Table 1).

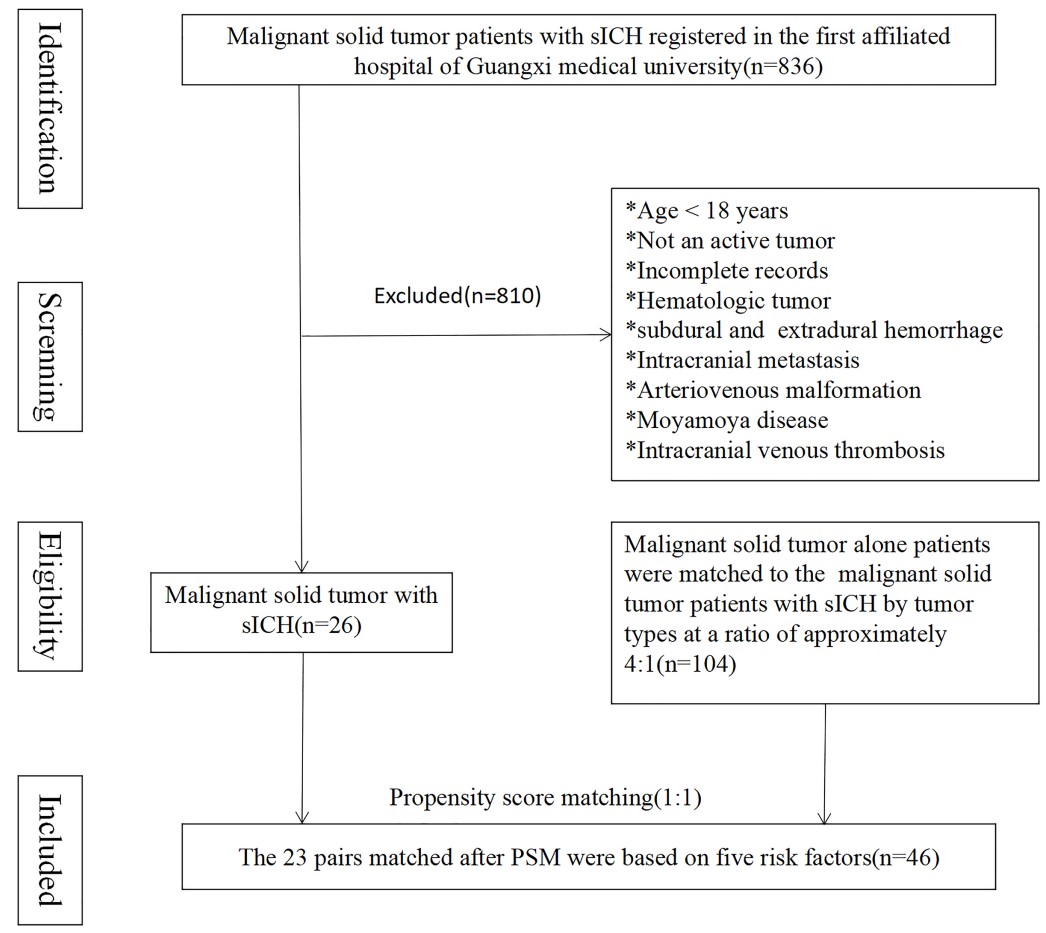

**Figure 1 The 23 pairs matched after PSM were based on five risk factors.**

## Laboratory examination data

Before PSM analysis, hematological parameters revealed no significant differences in white blood cell counts, RBC counts, and platelet counts were found between the MST-sICH and control groups, with $p$-values of 0.076, 0.057, and 0.161, respectively. However, hemoglobin (HGB) levels were significantly lower in the MST-sICH group than the control group ($p$ = 0.007). Concurrently, neutrophil counts (NCs), lymphocyte counts (LYCs), and the neutrophil-to-lymphocyte ratio (NLR) were significantly higher in MST-sICH patients than control patients, with $p$-values of 0.002, <0.001, and <0.001 respectively.

Liver function tests showedthat albumin levels were significantly higher in the MST-sICH group than the control group ($p$ = 0.012). However, no significant differences in gamma-glutamyl transferase, aspartate aminotransferase, alanine aminotransferase, the ratio of alanine aminotransferase to aspartate aminotransferase, and prealbumin were observed, with $p$-values of 0.035, 0.124, 0.793, 0.249, and 0.363, respectively.

Renal function tests indicated that urea and cystatin C levels were significantly higher in MST-sICH patients than control patients, with $p$-values of 0.027 and 0.003, respectively.

**Table 1  Demographics of MST-sICH group patients ($n = 26$).**

| Characteristic | All patients, $n = 26$ (%) | Favorable group, $n = 16$ (%) | Unfavorable group, $n = 10$ (%) |
|---|---|---|---|
| **Age (year)** | | | |
| 18–60 | 12 (46.2) | 6 (37.5) | 6 (60) |
| >60 | 14 (53.8) | 10 (62.5) | 4 (40) |
| Male | 20 (76.9) | 11 (68.8) | 9 (90) |
| **Risk factors** | | | |
| Hypertension | 10 (38.5) | 6 (37.5) | 4 (40) |
| Type 2 diabetes | 2 (7.7) | 2 (12.5) | 0 (0) |
| CHD | 1 (3.8) | 1 (6.3) | 0 (0) |
| DVT | 2 (7.7) | 2 (12.5) | 0 (0) |
| Anticoagulants | 2 (7.7) | 2 (12.5) | 0 (0) |
| Anti platelets | 2 (7.7) | 1 (6.3) | 1 (10.0) |
| **Oncological data** | | | |
| **Origin** | | | |
| Liver | 6 (23.1) | 2 (12.5) | 4 (40.0) |
| Bile duct | 1 (3.8) | 1 (6.3) | 0 (0) |
| Gastric | 2 (7.7) | 1 (6.3) | 1 (10.0) |
| Intestinal | 6 (23.1) | 3 (18.8) | 3 (30.0) |
| Breast | 1 (3.8) | 1 (6.3) | 0 (0) |
| Lung | 8 (30.8) | 4 (25.0) | 4 (40.0) |
| Paranephros | 1 (3.8) | 1 (6.3) | 0 (0) |
| Prostate | 1 (3.8) | 1 (6.3) | 0 (0) |
| **Metastasis** | | | |
| Liver | 4 (15.4) | 3 (18.8) | 1 (10.0) |
| Lung | 2 (7.7) | 1 (6.3) | 1 (10.0) |
| Others | 4 (15.4) | 4 (25) | 0 (0) |
| **pathological types** | | | |
| SCLC | 2 (7.7) | 1 (6.3) | 1 (10.0) |
| SCC | 3 (11.5) | 3 (18.8) | 0 (0) |
| HCC | 5 (19.2) | 2 (12.5) | 3 (30.0) |
| Adenocarcinoma | 16 (61.5) | 10 (62.5) | 6 (60.0) |
| **Treatment of malignancy[a]** | | | |
| Surgery | 11 (42.3) | 5 (31.3) | 6 (60) |
| chemotherapy | 6 (23.1) | 5 (31.3) | 1 (10) |
| Interventional | 3 (11.5) | 0 (0) | 3 (30) |
| untreated | 10 (38.5) | 7 (43.8) | 3 (30) |
| **Clinical** | | | |
| Symptom | | | |
| Coma | 10 (38.5) | 2 (12.5) | 8 (80) |
| Paralysis | 9 (34.6) | 4 (25) | 5 (50) |
| Aphasia | 4 (15.4) | 1 (6.25) | 3 (30) |
| Sensory | 1 (3.8) | 1 (6.25) | 0 |

(Continued)

| Characteristic | All patients, n = 26 (%) | Favorable group, n = 16 (%) | Unfavorable group, n = 10 (%) |
|---|---|---|---|
| Headache | 6 (23.1) | 4 (25) | 1 (10) |
| Nausea | 8 (30.8) | 4 (25) | 4 (40) |
| **Hematoma volume ≥ 32ml[b]** | 9 (34.6) | 2 (12.5) | 7 (70) |
| **Hematoma location** | | | |
| Frontal lobe | 7 (26.9) | 1 (6.3) | 6 (60) |
| Temporal lobe | 5 (19.2) | 1 (6.3) | 4 (40) |
| Parietal lobe | 13 (50.0) | 6 (37.5) | 7 (70) |
| Occipital lobe | 5 (19.2) | 2 (12.5) | 3 (30) |
| Basal ganglia | 5 (19.2) | 4 (25) | 1 (10) |
| cerebellum | 4 (15.4) | 4 (25) | 0 |
| thalamus | 2 (7.7) | 2 (12.5) | 0 |
| **sICH after surgery** | 5 (19.2) | 1 (6.25) | 4 (40) |
| **Treatment of sICH** | | | |
| surgery | 6 (23.1) | 3 (18.8) | 3 (30) |
| conservative | 20 (76.9) | 13 (81.3) | 7 (70) |
| **Prognosis** | | | |
| In-hospital mortality | 10 (38.5) | 0 (0) | 10 (100) |

Notes:
Favorable group: mRS ≤ 2; unfavorable group: mRS > 3.
[a] Some patients receive multiple treatments.
[b] Optimal cut-off value was determined by the Youden index, convert a continuous variable into a categorical variable.
Abbreviations: CHD, chronic heart disease, DVT, deep vein thrombosis, SCLC, small cell lung cancer, SCC, squamous cell carcinoma, HCC, hepatocellular carcinoma.

Furthermore, the creatinine clearance rate was significantly lower in the MST-sICH group than the control group ($p = 0.008$).

After PSM analysis, HGB levels and LYCs were found to be significantly lower in the MST-sICH group than the control group, with $p$-values of 0.020 and <0.001, respectively. Conversely, NCs and NLR were significantly higher in MST-sICH patients than control patients, with $p$-values of 0.042 and <0.001, respectively (Table 2).

## Multivariate logistic regression analysis of predictors for MST-sICH

Multivariate logistic regression analysis was conducted to identify the risk factors for sICH in patients with malignant solid tumors. Only variables achieving a significance threshold of $p < 0.05$ were retained for inclusion in the multivariate model. The analysis identified three independent predictors of sICH. Lower HGB levels were associated with an increased risk of sICH (odds ratio (OR): 0.959, 95% confidence interval (CI) [0.928–0.992]). Similarly, elevated LYCs significantly predicted the occurrence of sICH (OR: 0.095, 95% CI [0.023–0.392]). Furthermore, an increased NLR was also a significant predictor of sICH (OR: 2.137, 95% CI [1.427–3.200]). These findings suggest that specific hematological parameters may serve as valuable markers for identifying patients at a heightened risk of developing sICH (Table 3).

**Table 2 Clinical and laboratory profiles of patients before and after PSM.**

| Variable | Before propensity score matching | | | After propensity score matching | | |
|---|---|---|---|---|---|---|
| | MST-sICH ($n$ = 26) | Control ($n$ = 104) | $p$-value | MST-sICH ($n$ = 23) | Control ($n$ = 23) | $p$-value |
| Age (year) | 61.8 ± 13.6 | 56.7 ± 12.3 | 0.061[a] | 59.2 ± 12.0 | 57.9 ± 12.2 | 0.726[a] |
| Hypertension | 10 (38.5) | 21 (20.2) | 0.051[c] | 8 (34.8) | 5 (21.7) | 0.326[c] |
| Type 2 diabetes | 2 (7.7) | 3 (2.9) | 0.254[c] | 1 (4.3) | 0 (0) | 0.312[c] |
| Current smoker | 9 (34.6) | 34 (32.7) | 0.852[c] | 8 (34.8) | 12 (52.2) | 0.234[c] |
| Drinking | 10 (38.5) | 28 (26.9) | 0.250[c] | 9 (39.1) | 14 (68.1) | 0.140[c] |
| Blood tests | | | | | | |
| WBC ($10^9$/L) | 7.7 (5.9–10.4) | 6.2 (5.2–8.4) | 0.076[b] | 7.9 (5.9–10.3) | 5.9 (4.9–8.6) | 0.236[e] |
| RBC ($10^{12}$/L) | 4.1(3.4–4.5) | 4.3 (3.9–4.7) | 0.057[b] | 4.0 (3.4–4.4) | 4.2 (4.0–4.7) | 0.083[e] |
| HGB (g/L) | 113.0 (92.3–128.7) | 128.0 (115.1–137.7) | **0.007**[b] | 108.0 ± 28.2 | 127.1 ± 14.7 | **0.020**[d] |
| PLT ($10^9$/L) | 228.8 ± 100.2 | 255.6 ± 83.1 | 0.161[a] | 232.3 ± 103.4 | 228.9 ± 117.7 | 0.916[d] |
| NCs ($10^9$/L) | 5.2 (4.0–8.8) | 3.5 (2.7–4.9) | **0.002**[b] | 5.4 (4.0–8.6) | 3.4 (2.4–4.1) | **0.042**[e] |
| LYCs ($10^9$/L) | 1.1 ± 0.5 | 1.9 ± 0.6 | **<0.001**[a] | 1.1 ± 0.6 | 1.9 ± 0.6 | **<0.001**[d] |
| NLR | 7.7 (5.9–10.7) | 1.9 (1.5–2.9) | **<0.001**[b] | 7.9 (6.0–10.6) | 1.9 (1.4–3.2) | **<0.001**[e] |
| PT (S) | 11.6 (10.2–13.2) | 11.0 (10.2–11.8) | 0.111[b] | 12.0 ± 1.9 | 11.8 ± 1.2 | 0.823[d] |
| INR | 1.0 (1.0–1.1) | 1.0 (0.9–1.0) | 0.114[b] | 1.0 ± 0.2 | 1.0 ± 0.1 | 0.749[d] |
| ATPP (S) | 30.6 ± 4.1 | 31.6 ± 3.3 | 0.190[a] | 30.7 ± 3.9 | 31.7 ± 2.4 | 0.314[d] |
| TT (S) | 12.0 (11.2–13.0) | 11.9 (11.2–12.8) | 0.744[b] | 12.1 ± 1.6 | 11.9 ± 1.2 | 0.533[d] |
| PA (%) | 99.5 ± 25.6 | 107.5 ± 22.4 | 0.113[a] | 96.7 ± 25.4 | 95.0 ± 17.7 | 0.825[d] |
| TP (g/L) | 66.8 ± 8.4 | 66.7 ± 6.6 | 0.967[a] | 66.3 ± 8.8 | 66.8 ± 4.7 | 0.807[d] |
| ALB (g/L) | 36.7 ± 5.6 | 39.5 ± 4.9 | **0.012**[a] | 36.7 ± 5.9 | 39.8 ± 3.8 | 0.053[d] |
| GGT (U/L) | 52.5 (24.8–152.3) | 33.5 (16.3–52.1) | 0.035[b] | 67.0 (25.0–159.0) | 44.0 (26.0–76.0) | 0.242[e] |
| AST (U/L) | 27.0 (21.8–43.0) | 23.5 (19.0–31.0) | 0.124[b] | 27.0 (22.0–49.0) | 29.0 (20.0–45.0) | 0.951[e] |
| ALT(U/L) | 22.0 (13.0–32.8) | 20.0 (13.0–34.3) | 0.793[b] | 23.0 (14.0–35.0) | 26.0 (15.0–52.0) | 0.301[e] |
| ALT/AST | 1.3 (1.0–2.3) | 1.2 (0.9–1.8) | 0.249[b] | 1.2 (1.0–2.3) | 1.0 (0.8–2.0) | 0.212[e] |
| Preablumin (mg/L) | 201.2 ± 93.0 | 217.2 ± 76.5 | 0.363[a] | 202.4 ± 98.9 | 195.2 ± 79.5 | 0.775[d] |
| Urea (μmol/L) | 5.9 ± 2.2 | 4.8 ± 1.5 | **0.027**[a] | 5.1 (4.4–8.1) | 4.7 (3.9–6.4) | 0.224[e] |
| Cr (μmol/L) | 82.5 ± 25.3 | 72.1 ± 14.9 | 0.054[a] | 84.4 ± 30.9 | 76.3 ± 11.3 | 0.311[d] |
| UA (μmol/L) | 297.5 (232.3–399.8) | 288.5 (241.8–354.7) | 0.639[b] | 316.1 ± 134.8 | 297.0 ± 74.4 | 0.549[d] |
| CCR (ml/min) | 85.3 ± 32.0 | 101.4 ± 26.1 | **0.008**[a] | 84.4 ± 30.9 | 95.3 ± 20.9 | 0.186[d] |
| Cys-C (mg/L) | 1.0 (0.8–1.2) | 0.8 (0.7–1.0) | **0.003**[b] | 1.0 (0.8–1.2) | 0.8 (0.7–1.0) | 0.073[e] |

**Notes:**

Values are expressed as the mean ± SD, median (±) or $n$ (%); percentages are rounded to the nearest decimal point and thus may not add up to 100. Bold values indicate $p$ < 0.05, representing statistical significance.

[a] Two independent samples t-test.
[b] Mann–Whitney U-test.
[c] Chi-square test or Fisher's exact test.
[d] Paired Samples Test.
[e] Wilcoxon signed-rank test.

Abbreviations: sICH, spontaneous intracerebral hemorrhage; WBC, white blood cell; RGB, red blood cell; HGB, hemoglobin; PLT, platelet; NCs, neutrophil counts; LYCs, lymphocyte counts; NLR, neutrophil to lymphocyte ratio; HCT, hematocrit; PT, prothrombin time; INR, international normalized ratio; APTT, activated partial thromboplastin time; TT, thrombin time; PTA, prothrombin activity; TP, total protein; ALB, albumin; GGT, gamma glutamine transpeptidase; AST, aspartate aminotransferase; ALT, alanine transaminase; ALT/AST, alanine transaminase to aspartate aminotransferase; Cr, creatinine; UA, uric acid; CCR, creatinine clearance rate; Cys-C, cystatin C.

**Table 3 Multivariate logistic regression analysis.**

| Factors | OR | 95%Cl | P-value |
|---------|-----|-------|---------|
| HGB | 0.959 | [0.928–0.992] | 0.014* |
| NCs | 1.148 | [0.958–1.375] | 0.135 |
| LYCs | 0.095 | [0.023–0.392] | 0.001* |
| NLR | 2.137 | [1.427–3.200] | <0.001* |

**Notes:**
* $p < 0.05$.
Abbreviations: OR, odds ratio; Cl, confidence interval.

**Table 4 Receiver operating characteristic (ROC) curve of analysis of predictors.**

| Variables | AUC | SE | p-value | 95% CI | Cut-off Value | Sensitivity | Specificity | Youden index |
|-----------|-----|-----|---------|--------|---------------|-------------|-------------|--------------|
| HGB (A) | 0.711 | 0.079 | 0.014 | [0.557–0.865] | ≤110.95 | 0.522 | 0.913 | 0.435 |
| LYCs (B) | 0.827 | 0.061 | 0.000 | [0.707–0.947] | ≤0.369 | 0.913 | 0.652 | 0.565 |
| NLR (C) | 0.938 | 0.037 | 0.000 | [0.866–1.000] | >3.673 | 0.957 | 0.870 | 0.827 |
| A+B | 0.888 | 0.048 | 0.000 | [0.795–0.982] | >0.389 | 0.870 | 0.783 | 0.653 |
| B+C | 0.958 | 0.026 | 0.000 | [0.908–1.000] | >0.3189 | 1.000 | 0.826 | 0.826 |
| A+C | 0.947 | 0.032 | 0.000 | [0.884–1.000] | >0.3190 | 0.957 | 0.870 | 0.827 |
| A+B+C | 0.955 | 0.027 | 0.000 | [0.901–1.000] | >0.269 | 1.000 | 0.826 | 0.826 |

**Note:**
Abbreviations: AUC, area under the curve; SE, standard error; Cl, confidence interval.

## Receiver operating characteristic (ROC) curve analysis

ROC was performed to evaluate the predictive value of HGB, LYCs and NLR individually and in combination for sICH in patients with malignant solid tumors. Among the individual markers, NLR exhibited the highest predictive accuracy (AUC: 0.938, 95% CI [0.866–1.000], sensitivity: 95.7%, specificity: 87.0%, Youden index: 0.827). LYCs and HGB also showed predictive value, with AUCs of 0.827 (95% CI [0.707–0.947], sensitivity: 91.3%, specificity: 65.2%, Youden index: 0.565) and 0.711 (95% CI [0.557–0.865], sensitivity: 52.2%, specificity: 91.3%, Youden index: 0.435), respectively.

Combined models improved the predictive performance. The combination of LYCs and NLR yielded an AUC of 0.958 (95% CI [0.908–1.000], sensitivity: 100%, specificity: 82.6%, Youden index: 0.826) and HGB with NLR had an AUC of 0.947 (95% CI [0.884–1.000], sensitivity: 95.7%, specificity: 87.0%, Youden index: 0.827). The three-factor model including HGB, LYCs and NLR achieved the highest overall accuracy (AUC: 0.955, 95% CI [0.901–1.000], sensitivity: 100%, specificity: 82.6%, Youden index: 0.826), indicating their robust potential for sICH prediction in this patient population (Table 4).

## DISCUSSION

Previous studies have indicated that both diagnosed and undiagnosed cancers are correlated with an increased likelihood of experiencing a stroke. The risk of suffering from ischemic and hemorrhagic strokes was found to be almost twice as high in individuals with unidentified cancer in the year preceding their diagnosis than in the general population, highlighting the significance of recognizing stroke susceptibility in patients with

undisclosed cancer and underscoring the clinical imperative of identifying high-risk individuals (*Andersen & Olsen, 2018*). Indeed, sICH may appear as the first symptom of advanced or distant metastases in patients with malignant tumors. The incidence of sICH is higher than that of ischemic stroke (*Zöller et al., 2012*; *Graus, Rogers & Posner, 1985*; *Navi et al., 2010*). However, since the clinical manifestations of MST-sICH closely resemble those of traditional sICH, it is crucial to identify the distinguishing features of MST-sICH in order to facilitate early detection of malignancies and enable personalized treatment choices. In the current study, we found that conventional high-risk factors like age, HTN, Type 2 diabetes, smoking status, and alcohol consumption may not represent the primary etiologies for sICH, consistent with previous studies (*Navi et al., 2010*; *Rogers, 2004*). These findings indicate that the etiology of MST-sICH may diverge from that of conventional sICH, suggesting potentially distinct disease mechanisms. Hence, we conducted an analysis of the primary site, pathological types, blood tests, and other diagnostic modalities to elucidate the etiology of MST-sICH. We found that decreased HGB levels and an increase in LYCs and NLR were potential independent risk factors for MST-sICH.

Anemia is a common occurrence in malignant tumor patients, with a prevalence of over 40%. This number may increase to up to 90% in individuals receiving chemotherapy (*Knight, Wade & Balducci, 2004*; *Tas et al., 2002*). Chemotherapy-induced anemia, resulting from the myelosuppressive effects of chemotherapy on RBC, production, is a significant contributor to low HGB observed in cancer patients (*Rodgers et al., 2012*). The etiological factors of anemia can be classified into different categories: excessive bleeding, increased breakdown of RBCs, and decreased production of RBCs (*Adamson, 2008*). Based on these three fundamental mechanisms, the manifestation of anemia in individuals with malignant neoplasms can exhibit considerable variability such as coagulation dysfunction, hemolysis, renal insufficiency, nutritional disorders, or inflammatory diseases. At the same time, the malignant tumor itself can affect iron chelation through bone marrow infiltration or inhibition of cytokines, reducing erythrocyte production and aggravating anemia (*Adamson, 2008*; *Steensma, 2008*). For example, anemia caused by bone marrow infiltration or inflammatory cytokines may lead to additional systemic changes that affect hemostasis and vascular health, further complicating the relationship between HGB levels and bleeding risk in these patients. Anemia in patients with malignant tumors is frequently correlated with venous thrombosis (VTE). Furthermore, the impact of anemia on VTE patients, and the risk of severe bleeding in VTE patients with mild to moderate/severe anemia reportedly increases proportionally with the severity of anemia (*Steensma, 2008*). However, in the present study, no significant differences in coagulation function were observed between the two groups, and there were few diagnosed cases of VTE. This may be attributed to other factors such as anemia itself, which could lead to a tendency to a bleeding tendency independent of traditional coagulation parameters. Both VTE and bleeding tendency are associated with alterations in platelet function. Anemia can impact the interaction between platelets and sites of vascular injury through hemodynamic changes, thereby influencing coagulation function (*Weisel & Litvinov, 2019*). The reduction in RBCs due to anemia significantly impacts the hemostatic function of platelets.

First, RBCs exert mechanical force on platelets, enhancing their ability to adhere to damaged blood vessel walls and increasing their stickiness. In addition, RBCs serve as a crucial source of phosphatidylserine, which facilitates thrombin production. Consequently, RBCs enhance thrombin production, thereby promoting platelet activation, fibrin formation, and clot stability (*Weisel & Litvinov, 2019*; *Whelihan et al., 2012*). Thus, an abnormal decrease in HGB levels, as frequently seen in chemotherapy-induced or tumor-related anemia, may inhibit platelet activation and impair coagulation function, potentially contributing to the mechanism of MST-sICH (*Carlisle et al., 2011*).

In the present study, a significant decrease in RBC counts was not observed nor was a significant difference in RBC counts found between the two groups. However, a significant decrease in HGB was observed in MST-sICH patients. The results indicate that although the number of RBCs remains relatively stable, there is a notable reduction in HGB levels. This suggests that the efficacy of the red blood cells in transporting oxygen may be compromised, potentially due to factors such as iron deficiency or other quality-related issues, rather than a mere reduction in their quantity (*Choi et al., 2024*). Based on previous studies on anemia in individuals with malignant tumors, our conclusion is that this presents as iron-deficiency anemia secondary to malignancy (*Lopez et al., 2016*). Previous studies have indicated that lower HGB levels upon admission are associated with an elevated risk of hematoma expansion and a poorer prognosis in terms of the impact of HGB on sICH (*Roh et al., 2019*). A reduction in HGB levels may result in a decrease in blood viscosity and oxygen delivery to tissues. This could lead to an exacerbation of hypoxic stress on blood vessels, potentially leading to a weakening of the vessel walls and an increased susceptibility to hemorrhage (*Hightower et al., 2012*; *Ugurel et al., 2019*). Thus, HGB may exert an influence on the coagulation process in some capacity. Thromboelastography of patients with sICH has revealed that, even in the absence of identifiable risk factors for coagulation dysfunction, relative coagulation dysfunction may still be present, which may contribute to the hypocoagulation of the patient's blood, leading to the occult onset of sICH (*Roh et al., 2023*). Furthermore, the reduction in HGB levels leads to alterations in blood composition. Furthermore, reduced HGB levels can impact the bioavailability of nitric oxide (NO), which plays a complex role in both tumor progression and vascular regulation (*Moore, Tymvios & Emerson, 2010*).

In cancer patients, the level of nitric oxide (NO) in the bloodstream plays a crucial role in tumor development. Cancer cells have the ability to produce NO independently, which subsequently activates signaling pathways linked to cell proliferation, angiogenesis, and metabolism. The autonomous secretion of NO by cancer cells further enhances their aggressive behavior and growth (*Liu et al., 2021*). This excessive NO production may not only support tumor growth but also disturb normal vascular function, contributing to an increased risk of hemorrhage (*Jamieson, 2017*). NO is also implicated in the vasodilation of cerebral blood vessels. Changes in cerebral blood flow can result in alterations in shear force on the endothelial wall of blood vessels. Upon sensing these changes, the endothelium of the blood vessels will modulate NO levels to induce relaxation of vascular smooth muscle, ultimately leading to vasodilation. In the event of decreased HGB levels, incomplete binding with NO could potentially result in an elevated levels of NO within the

bloodstream, leading to abnormal and sustained dilation of cerebral blood vessels. The chronic vasodilation, when coupled with the weakened vessel walls resulting from hypoxic stress and other contributing factors, may elevate the probability of vessel rupture and haemorrhage. This process may offer a potential mechanism for the occurrence of MST-sICH (*Hoiland et al., 2022*).

It is also important to consider the impact of vascular physiological changes in the context of malignant tumors. Tumors have been observed to induce a pro-inflammatory and pro-thrombotic state in the body, which can further compromise vascular integrity (*DeNardo & Ruffell, 2019*). Excessive production of NO results in endothelial dysfunction and reduced capacity to clear oxidized low-density lipoproteins, subsequently contributing to the progression of atherosclerosis (*Gliozzi et al., 2019*). Atherosclerosis is intricately linked to the onset and progression of cancer, stemming from the expansion of modified cell clones within the framework of local tissue injury, inflammation, and genomic instability (*Markowitz, 1997*). During the early stages of atherosclerotic plaque development, there is a correlation between inflammation and inflammatory factors (such as chemokines and cytokines) with increased levels of growth factors and cell growth in endothelial cells, mononuclear inflammatory cells, macrophages, and smooth muscle cells (*Sullivan, Sarembock & Linden, 2000*). Similarly, inflammation has long been considered a major precursor to the development of cancer under both infected and non-infected conditions (*Ardestani et al., 1999*; *Morré et al., 2000*). The evidence suggests that the interaction between cancer and atherosclerosis may intensify vascular instability, thereby increasing the risk of sICH in cancer patients. This shared inflammatory pathway implies that the relationship between these two conditions may be more complex than previously thought (*Jirasek et al., 2016*; *Leca et al., 2023*). In the current study, abnormal inflammatory indexes were also observed in blood tests. For example, MST-sICH patients displayed elevated LYCs and NLR levels compared to malignant tumor patients without sICH, indicating a heightened inflammatory response within their circulatory system. Interestingly, NLR has been identified as a predictor of atherosclerosis, with elevated levels of NLR being linked to the prevalence of intracranial atherosclerosis (*Nam et al., 2018*). Previous studies have shown that during the development of atherosclerosis, the number of neutrophils in the blood increases. Neutrophils are attracted to the plaque to secrete pro-inflammatory products, such as elastase, myeloperoxidase, and reactive oxygen species, which may lead to damage to the blood vessel wall (*Erturk et al., 2014*). Atherosclerosis within the central nervous system blood supply has been correlated with deep brain hemorrhagic cerebrovascular disease, and this association becomes more pronounced as the severity of atherosclerosis increases (*Romero et al., 2016*). Persistent inflammation-induced damage to blood vessel walls may represent an additional potential mechanism for MST-sICH.

Although PSM was employed in the present study to control for conventional vascular risk factors, it is possible that unidentified confounding variables may have influenced the results. PSM effectively balanced the groups for the following established risk factors for cerebrovascular disease and cancer: age, hypertension, Type 2 diabetes, smoking, and alcohol consumption. However, additional patient characteristics, such as specific
oncological factors (*e.g.*, tumour stage, treatment regimen) and medication history, particularly the use of anticoagulants or antiplatelet agents, could also exert an influence on the likelihood of sICH in this cohort. Although we excluded patients with coagulopathy and intracranial metastasis to focus on solid tumours without direct neurological impact, the underlying heterogeneity within the cancer population, including varying degrees of systemic inflammation and immune response, could introduce additional variability in sICH risk that PSM could not entirely eliminate. Moreover, while our study identifies multiple independent risk factors, including HGB, LYCs, and NLR, their integration into a predictive model offers significant clinical benefits. The combined diagnostic model exhibited superior predictive accuracy in comparison to the individual factors. While NLR exhibited high predictive power (AUC: 0.938, sensitivity: 95.7%, specificity: 87.0%), integrating HGB, LYCs, and NLR into a single model resulted in an AUC of 0.955, with sensitivity and specificity values of 100% and 82.6%, respectively. This enhanced performance highlights the complementary role of these markers in capturing the multifaceted pathophysiological alterations associated with MST-sICH. HGB is indicative of anemia-related hypoxia and vascular changes, while LYCs and NLR provide insights into systemic inflammation and immune responses. The combination of these factors facilitates a more comprehensive assessment of MST-sICH risk, addressing its heterogeneous and multifactorial aetiology. The combined model has the potential to significantly enhance risk stratification and improve patient management. By identifying high-risk patients, the model could facilitate early interventions, closer monitoring and the implementation of individualised therapeutic strategies, with the aim of reducing the incidence and severity of MST-sICH.

Our study uncovers the contribution of malignant tumors to the development of MST-sICH by inducing persistent alterations in the internal environment that impact blood vessel clotting function and pathophysiological status. Our findings provide further insights into the concealed pathogenesis of MST-sICH. Nevertheless, our study is subject to certain limitations, including its retrospective design, single-center study setting, and inherent selection bias. Furthermore, due to the small sample size, we were unable to adequately evaluate the sensitivity and specificity of various risk factors or investigate the long-term prognosis of patients. These limitations may have influenced the generalizability and depth of our findings.

## CONCLUSION

In conclusion, this study has examined the detailed characteristics of MST-sICH and conducted an analysis of the risk factors associated with its occurrence. Our findings suggest that significantly reduced HGB levels and high NLR and LYCs may serve as independent risk factors for the development of MST-sICH in patients with malignant tumors; however, further research is necessary to validate our results.

### Funding

This work was supported by the National Natural Science Foundation of China (Grant numbers (82260243)), and the Guangxi Medical and Health Appropriate Technology Promotion Project (Grant numbers (S2021101). The funders had no role in study design, data collection and analysis, decision to publish, or preparation of the manuscript.

### Grant Disclosures

The following grant information was disclosed by the authors:
National Natural Science Foundation of China: 82260243.
Guangxi Medical and Health Appropriate Technology Promotion Project: S2021101.

### Competing Interests

The authors declare that they have no competing interests.

### Author Contributions

- Shuolin Liang conceived and designed the experiments, performed the experiments, prepared figures and/or tables, authored or reviewed drafts of the article, and approved the final draft.
- Liuyu Liu performed the experiments, prepared figures and/or tables, and approved the final draft.
- Bin Qin analyzed the data, prepared figures and/or tables, and approved the final draft.
- Shengri Chen analyzed the data, prepared figures and/or tables, and approved the final draft.
- Zhijian Liang conceived and designed the experiments, authored or reviewed drafts of the article, and approved the final draft.

### Human Ethics

The following information was supplied relating to ethical approvals (*i.e.*, approving body and any reference numbers):

Medical Ethics Committee of the First Affiliated Hospital of Guangxi Medical University (No. 2023-E246-01).

### Data Availability

Data can be found in the Supplemental Information.

### Supplemental Information

Supplemental information for this article can be found online at http://dx.doi.org/10.7717/peerj.18737#supplemental-information.

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
