# Peer review of "Malignant solid tumor-related spontaneous intracerebral hemorrhage: a propensity score matching study"

_PeerJ, doi:10.7717/peerj.18737_

## Round 0.1 · original submission · Major Revisions

First, the Abstract and Introduction need strengthening to better contextualize the research and clearly articulate its significance. The Abstract should include a concise background on sICH and its relationship with malignant solid tumors, while the Introduction requires a more compelling opening and clearer exposition of how this study addresses existing knowledge gaps.

The methodology section needs more detailed explanation of the propensity score matching process, including justification for the selected matching variables and evidence of matching quality. The exclusion criteria should be more precisely defined, and the imaging procedures need to be described in greater detail. The statistical analysis should be expanded to include comprehensive ROC analysis and consideration of additional variables such as cancer stage and gender effects.

The Results section would benefit from reorganization, particularly in the presentation of laboratory data, with clearer separation of pre- and post-PSM findings. Additional analyses comparing outcomes across different tumor types and examining gender-specific risks would strengthen the findings. Visual presentations of key results, including ROC curves, would enhance the manuscript's clarity.

The Discussion section requires substantial expansion to address potential confounding factors, provide deeper analysis of the anemia findings, and consider additional inflammatory markers beyond NLR. The clinical implications of your findings should be more thoroughly explored, including specific preventive strategies and treatment recommendations.
Please ensure all abbreviations are defined at first use and include the missing Tables 3 and 4.

Reviewer 1 ·

Basic reporting

Well written document without any concerns with the flow or language use. No Comment

Experimental design

No Comment. Good experimental design with good inclusion/exclusion criteria and methodlogy chosen.

Validity of the findings

No comment

Reviewer 2 ·

Basic reporting

1)The manuscript is generally well-written, presenting clear and professional language. However, there are instances of awkward phrasing and grammatical errors that could hinder comprehension. A thorough proofreading by a proficient English speaker or a professional editing service is recommended.
2)The data in the results section is presented clearly, but consider using charts to more visually present key findings, especially the results of the ROC curve analysis.
3)Make sure that all abbreviations are clearly defined when they first appear and remain consistent throughout the text.

Experimental design

4)The research question is well-defined and addresses a significant knowledge gap regarding the risk factors for sICH in patients with malignant solid tumors.
5)The methodology is generally rigorous, employing a propensity score matching (PSM) approach to control for confounding variables. The authors should provide more detail on the matching process, the specific variables considered for matching, and the statistical methods used to ensure replicability.
6)Specific variables like anticoagulant use and metastatic status were excluded from the analysis. It would be beneficial to clarify how matching variables were selected and the rationale behind the exclusion of certain confounders.

Validity of the findings

7)The results are presented clearly, and the authors appropriately link their findings to the original research question. However, the discussion could be strengthened by addressing potential confounding factors that may not have been fully controlled for in the analysis and whether there are any confounders that could not be fully accounted for by PSM.
8)Consideration of the types or causes of anemia (e.g., chemotherapy-induced anemia) that might influence hemoglobin (HGB) findings is necessary, especially given the identified link between low HGB levels and MST-sICH.
9)The study mentions that the neutrophil-to-lymphocyte ratio (NLR) is a predictor of sICH. Were other markers of inflammation considered in the analysis, such as C-reactive protein (CRP) or interleukin-6 (IL-6)?
10)It should be clarified whether the effect of different cancer stages on sICH risk was analyzed and if there were notable differences associated with cancer stage.
11)Did you observe differences in sICH prognosis between patients with different tumour types (e.g. lung, liver or gastrointestinal cancers)?
12)Did the study consider the potential role of anemia treatment, such as iron supplements or transfusions, in reducing the risk of sICH?
13)Did the study assess any potential differences in sICH risk between male and female cancer patients, and if not, do you think gender might play a role?

Additional comments

14)Overall, the manuscript presents valuable insights into the risk factors associated with sICH in patients with malignant solid tumors, with particular emphasis on findings related to hemoglobin levels, lymphocyte counts, and the neutrophil-to-lymphocyte ratio as potential risk factors.

Reviewer 3 ·

Basic reporting

This retrospective study investigated the clinical characteristics and risk factors for spontaneous intracerebral hemorrhage (sICH) in patients with active solid tumors. Using a 1:1 propensity score matching approach, the study found that decreased hemoglobin levels, increased lymphocyte counts, and an elevated neutrophil-to-lymphocyte ratio (NLR) were significantly associated with MST-sICH. Multivariate logistic regression analysis confirmed these associations, and an ROC curve analysis demonstrated the high predictive power of these factors for sICH in this patient population. The findings suggest that monitoring these hematological parameters could be valuable for identifying patients at increased risk of sICH, potentially enabling early intervention and preventive measures. However, there are still some issues that need to be addressed.
1. The Abstract could be strengthened by providing a brief background on sICH and its association with malignant solid tumors. This would help readers understand the significance of the study.
2. The conclusion in the Abstract could be strengthened by directly addressing the clinical implications of the findings. For example, it could mention how the identified risk factors can be used to identify high-risk patients and potentially guide preventive measures.
3. The introduction starts with general information about cerebrovascular disease and ICH, which is common knowledge. It lacks a strong hook or a compelling statement to grab the reader's attention and highlight the urgency or novelty of the research problem.
4. The introduction mainly describes existing research findings rather than critically analyzing them. It lacks a clear argument about how the current study fills a gap in the literature or addresses a specific limitation of previous research. In addition, the introduction mentions that the findings are expected to inform future therapeutic strategies, but it doesn't specifically outline how. A more detailed explanation of the potential implications and contributions of the study would enhance the impact of the introduction.
5. While the exclusion criteria are comprehensive, some are vague. For example, "incomplete medical records" could be more specific. Defining what constitutes an incomplete record would enhance clarity and reproducibility. The rationale for some exclusion criteria, such as "presence of liquid malignancy" and "occurrence of subarachnoid hemorrhage," could be clarified. Explain why these conditions were excluded and how they might influence the study findings.
6. Explain why age, HTN, diabetes, smoking status, and alcohol consumption were chosen as variables for the PSM. Discuss how these variables potentially influence the development of sICH and their relevance to the study's objective. It would be beneficial to mention whether balance diagnostics were conducted to assess the effectiveness of PSM in achieving comparable groups on key covariates after matching. This would ensure that confounding factors were adequately controlled.
7. The results in “Demographic and clinical characteristics” mainly present descriptive statistics without providing much interpretation or discussion of the potential implications of the findings. For example, while the section mentions coma, nausea, and paralysis as common clinical symptoms, it doesn't discuss the potential significance of these symptoms in relation to sICH in cancer patients.
8. The section in “ Imaging data” mentions that "traditional risk factors for sICH and intracranial tumor metastasis were excluded" through cerebrovascular imaging evaluations. However, it doesn't provide any details about the specific imaging techniques used or the criteria for excluding these factors. A more detailed explanation of the imaging procedures would enhance the clarity and reproducibility of the study. In addition, the section doesn't mention the imaging characteristics of the hematomas, such as size, shape, or presence of associated features (e.g., edema, mass effect). This information could provide valuable insights into the nature and extent of the hemorrhages.
9. The presentation of the data in “Laboratory examination data” is somewhat fragmented, making it difficult to follow the key findings. Organizing the results based on specific categories (e.g., hematological parameters, liver function, renal function) and then presenting the results before and after PSM analysis separately would enhance readability.
10. The section in “Receiver operating characteristic (ROC) curve analysis” doesn't provide information about the performance of the combined diagnostic model that included all three factors (HGB, LYCs, and NLR). It simply mentions that NLR maintained a high predictive value in this model but doesn't offer any details about the model's overall accuracy or sensitivity/specificity.
The Discussion section needs to be significantly revised to strengthen its argument, provide deeper interpretation of the findings, and address limitations more thoroughly. By focusing on the study's specific contributions, addressing the research question directly, and providing a more nuanced and critical interpretation of the results, the discussion section can be made more impactful and informative.
11. Some sentences are lengthy and could be rephrased for improved clarity and readability. For example, the sentence describing routine blood examination parameters could be broken down into multiple shorter sentences. In addition, The article lacks Tables 3 and 4.

Experimental design

_

Validity of the findings

_

---

## Round 0.2 · accepted · Accept

Authors have addressed all comments. I have no more questions.
This paper can be accepted for publication.

Reviewer 2 ·

Basic reporting

No comment.

Experimental design

No comment.

Validity of the findings

No comment.

Additional comments

The quality of the article has greatly improved through the author's revisions.

Reviewer 3 ·

Basic reporting

The author's revisions are constructive and substantially improve the deficiencies of the manuscript, basically address my concerns, and the current version can be accepted.

Experimental design

no comment

Validity of the findings

no comment